# Effect of Leaf Surface Regulation of Zinc Fertilizer on Absorption of Cadmium, Plumbum and Zinc in Rice (*Oryza sativa* L.)

**Jingyi Hu, Ronghao Tao, Chi Cao, Junhao Xie, Yuxin Gao, Hongxiang Hu, Zhongwen Ma \* and Youhua Ma \***

Anhui Province Key Laboratory of Farmland Ecological Conservation and Pollution Prevention, College of Resources and Environment, Anhui Agricultural University, Hefei 230036, China
\* Correspondence: mazhongwen@ahau.edu.cn (Z.M.); yhma@ahau.edu.cn (Y.M.)

**Abstract:** The accumulation of heavy metals in rice is bound to affect human health and safety. In order to ensure food security, this study explores the effect of leaf surface regulation of zinc (Zn) fertilizer on the safety of rice in cadmium (Cd)-plumbum (Pb) polluted farmland. Through field experiments, the leaf surface control treatment of zinc fertilizer was set up, and the effects of leaf surface control of zinc fertilizer on rice yield, Cd and Pb concentration in different parts and zinc, nitrogen, phosphorus and potassium concentration in brown rice were studied in the growing period of rice. The results showed as follows: (1) Spraying twice or more in the growth stage of rice could increase the yield by 6.77–7.29% compared with the blank, which was significantly higher than that of single spraying in a certain growth stage. (2) After spraying zinc fertilizer on the leaf surface, Cd and Pb concentration in brown rice decreased by 29.52–56.01% and 11.10–28.34%, respectively, compared with CK. Two or more times of spraying can make Cd concentration in brown rice meet the Chinese standard GB 2762-2022, and one time of spraying can make Pb concentration in brown rice meet the standard. (3) Leaf surface control of zinc fertilizer could reduce the Cd enrichment ability of rice plant parts, and the Cd enrichment coefficient of brown rice was significantly reduced by 28.18–55.02%. Leaf surface control of zinc fertilizer can reduce Cd and Pb concentration in brown rice by reducing the transport ability of heavy metal Cd and Pb in rice roots to straw and then to brown rice. (4) The zinc concentration in brown rice was 18.16–20.68 mg·kg$^{-1}$, which was 18.21–34.64% higher than that in CK, and the zinc enrichment effect was the most significant after spraying three times. Meanwhile, the nitrogen, phosphorus and potassium concentration in brown rice also increased with the increase of spraying times. By comprehensive analysis, the leaf surface control of zinc fertilizer can reduce the Cd and Pb concentration in rice, and the Cd and Pb concentration in brown rice can meet the Chinese standard GB 2762-2022 by spraying twice. At the same time, it can improve the concentration of zinc, nitrogen, phosphorus and potassium in brown rice, is feasible and has high economic benefits.

**Keywords:** rice; foliar zinc fertilizer; cadmium and plumbum pollution; heavy metals; farmland

## 1. Introduction

Rice (*Oryza sativa* L.) is the largest grain crop in China and has strong enrichment capacity for heavy metals. Rice has a strong adsorption capacity for heavy metals, and the heavy metal content of rice is higher than that of other dry crops. According to the National Soil Pollution Survey Bulletin released by China in 2014, 19.4% of China's arable land exceeded the standard for heavy metals. The excessive concentration of heavy metals in soil can increase the accumulation of heavy metals in crops and seriously affect human health and safety. The survey found that about 10% of rice produced in China each year contained excessive levels of heavy metals. As heavy metals such as Cd and Pb are non-essential elements of the human body and have strong toxicity to organisms, heavy metals accumulated in crops can be enriched in the human body along with the transmission of the food chain, causing potential harm to human health [1]. Unlike organic pollutants, heavy metals are not degradable and can pose risks to human health directly or indirectly through

the food chain [2]. Tsukahara et al. [3] found through investigation that eating rice with excessive cadmium is the main source of cadmium in the human body. Long-term, low-dose cadmium exposure caused by consumption of cadmium rice is carcinogenic, teratogenic and mutagenic, posing a serious threat to food production security and human health. Pb can weaken the body's immune system in a variety of organisms, including humans, and cause a range of symptoms including dizziness, headaches and memory loss. In recent years, the technical measures of resistance and control of heavy metal accumulation in polluted farmland rice have become a research hotspot, especially leaf resistance and control technology, which can inhibit the transfer of heavy metals to rice grains and reduce the heavy metal concentration in rice by spraying conditioner at an appropriate time. Compared with the common soil passivation technology, it has advantages of stable effect, low cost and environmental friendliness. It was found that foliar spraying of mineral elements [4,5] could effectively inhibit the transport of heavy metals from rice to grain and reduce the concentration of heavy metals in rice. The vegetative organs of rice also have the ability to inhibit heavy metal transport, which can fix a large amount of heavy metals in the cell wall of the vegetative body or in vacuoles, thus inhibiting the transport of heavy metals to the grain. The vegetative organs with the highest concentration of heavy metals are the roots and nodes of rice [6]. At the same time, rice cells can also identify essential and harmful elements, and preferentially transport nutrients and mineral elements necessary for growth [7].

Agronomic measures such as the appropriate addition of zinc fertilizer are considered to be the most economical, convenient and effective safe utilization technology to reduce Cd absorption and accumulation by crops [8,9]. Studies have shown that zinc application can reduce the absorption, transfer and accumulation of cadmium in rice. Zn has synergistic and antagonistic effects on the absorption and transport of Cd in monocotyledon plants such as rice [10,11]. Lv et al. [12] found that in acid Cd-polluted paddy soil, leaf spraying with 0.5% $ZnSO_4$ could reduce the Cd concentration in rice grains by 19.03–32.55%. Duan and Han's study also showed that leaf zinc fertilizer could reduce Cd concentration in rice grains [13,14]. Other studies have shown that zinc fertilizer spraying can reduce Cd concentration in rice grains by inhibiting Cd absorption by rice roots and Cd transport by rice plants [10,12]. However, some studies have shown a synergistic effect between Zn and Cd in monocotyledon plants. For example, exogenous Zn increases the accumulation and transport of Cd in rice soils with moderate Cd contamination [15,16]. Zn can affect Cd accumulation by regulating Cd uptake by roots and root-shoot transport of crops [17]. Lv et al. [16] found that foliar application of Zn reduced the concentration of Cd in early rice grains by inhibiting the root absorption of Cd and its subsequent transport from root and flag leaf to the first node and from ear to grain. Other studies have reported that the absorption, transport and distribution of Cd in rice are strongly influenced by rice varieties, growing seasons and other factors [18,19].

Zinc is an essential trace element for the growth and development of human beings and crops, and plays an important role in maintaining physiological functions [20–24]. Zinc exists in various organs of the human body and is involved in the synthesis of a variety of enzymes, DNA and RNA, playing a role in maintaining the nervous system, immune system and regulating the secretion of reproductive hormones [25,26]. Zinc deficiency caused by inadequate dietary intake is one of the most common problems in developing countries, resulting in human immune dysfunction, growth retardation and cognitive impairment [27]. Genetic biofortification and agricultural biofortification are the key to zinc fortification of food crops [28]. Although plant breeding can provide sustainable solutions, it is usually tested in controlled laboratories or greenhouses where trace elements are abundant, and its effect in the case of actual zinc deficiency is unknown [29]. In contrast, the application of soil and leaf zinc fertilizer under field conditions is more conducive to crop biological reinforcement [30]. In the main wheat producing areas mainly dominated by zinc deficiency and potential zinc deficiency, soil zinc fertilization has not obviously improved zinc concentration or yield in seeds [31]. In addition, high pH and a high

concentration of calcium carbonate in calcareous topsoil can reduce the bioavailability of zinc application and the absorption of zinc by plant roots [32]. Based on this, the utilization rate of soil zinc fertilizer in this type of soil is very low. Increasing the application amount of zinc fertilizer will not only not improve crop yield, but also cause waste of zinc fertilizer. Foliar spraying becomes a good option when zinc deficiency cannot be alleviated by soil application. The time of foliar spraying of trace elements is one of the important factors that determine crop yield and quality. Ozturket et al. [33] found that foliar zinc application could significantly increase grain zinc concentration in wheat.

In some areas of southern China, Cd-Pb complex-contaminated farmland exists. Previous studies have shown that leaf surface regulation has an effect on rice grown in single polluted farmland, but the effect on rice grown in Cd-Pb complex-polluted farmland is unknown. In addition, studies on reducing heavy metals while improving nutrients such as zinc in rice are rarely reported. In this study, through field experiments, the effects of zinc foliar fertilizer spraying at different growth stages on Cd and Pb absorption of rice and on the improvement of zinc and other nutrients in brown rice were discussed, in order to provide theoretical and practical basis for the remediation of Cd and Pb compound polluted farmland and the safe utilization of agricultural products.

## 2. Materials and Methods

### 2.1. Test Site

The experimental site was Cd and Pb compound-polluted farmland in Yi 'an District, Tongling City, Anhui Province. The soil pH value was 5.35, the total concentration of Cd in soil was 1.53 mg·kg$^{-1}$, higher than the risk control value of soil pollution in agricultural land, and the DTPA-Cd was 0.682 mg·kg$^{-1}$. The total concentration of Pb was 82.67 mg·kg$^{-1}$, which was higher than the screening value of soil pollution risk of agricultural land, and the DTPA-Pb was 36.85 mg·kg$^{-1}$. Soil organic matter (OM) concentration was 18.65 g·kg$^{-1}$, total nitrogen concentration was 0.79 g·kg$^{-1}$, alkali-hydrolyzed nitrogen concentration was 77.69 mg·kg$^{-1}$, available phosphorus concentration was 17.65 mg·kg$^{-1}$ and available potassium concentration was 142.38 mg·kg$^{-1}$.

### 2.2. Experimental Treatments

CK: tiller stage–heading stage–filling stage
T1: heading stage was sprayed with leaf zinc fertilizer once
T2: filling stage was sprayed with leaf zinc fertilizer once
T3: heading stage–filling stage was sprayed with leaf zinc fertilizer twice
T4: Leaf zinc fertilizer was sprayed 3 times from tiller stage to heading stage to grouting stage

Experimental materials: The rice variety tested was Guizhao 2, which was suitable for planting in the local area. Leaf zinc fertilizer is fluid zinc fertilizer (Zn $\geq$ 150 g·L$^{-1}$, pH: 6–7), which is produced by Anhui Luokuo Zhifeng Agricultural Technology Co., LTD, Hefei China. The concentration of foliar fertilizer was 0.75 kg·hm$^{-2}$ each time, and 600 kg·hm$^{-2}$ was added with water and diluted 1000 times. After each treatment, the spraying was finished before 8 a.m. on a sunny day, and the front and back sides of the foliar and the ear were evenly sprayed. It was better to apply the liquid drops on the leaf and ear.

The experimental design of the blocks adopted random distribution, or a total of 5 treatment blocks, each treatment block set with 3 repetitions, for a total of 15 treatment blocks, with each block 20 m$^2$ in area, and each block with a cement protection ridge, clean water irrigation and cut off pollution sources. According to the local high yield cultivation technique, the base fertilizer was 45% (15-15-15) compound fertilizer 600 kg·hm$^{-2}$, which was applied 1–2d before transplanting, and the rice density was 13 cm × 30 cm. After 20 days, 150 kg·hm$^{-2}$ urea was applied as tillering fertilizer, 112.5 kg·hm$^{-2}$ potassium fertilizer was applied and 112.5 kg·hm$^{-2}$ urea was applied as booting fertilizer at the booting stage.

### 2.3. Sample Collection

The samples were collected at the mature stage of rice on 27 September 2021 and the actual yield was measured. Five rice plants were selected from each plot according to the plum blossom sampling method. The soil samples were collected on the day of rice sample collection, and the soil (0–20 cm) in the root zone of rice was directly collected at the point of rice sample collection. The soil samples were composed of mixed soil samples, weighing about 1500 g. The rice plants were cleaned with tap water and deionized water, and then the whole plant was divided into roots, stems, leaves and rice grains. The plants were defoliated at 105 °C for 30 min and dried at 80 °C to constant weight. After drying, the rice is roughened according to the industry standard NY/T 83-2017 in China, and the brown rice and husks are separated. The dry weight of each part is weighed and crushed by stainless steel mill. After air drying in a cool place, the soil samples were crushed and ground with 10-mesh sieve and 100-mesh sieve into a Ziplock bag for reserve.

### 2.4. Sample Determination

The determination of many elements in rice samples refer to the GB 5009.268-2016 in China. The Jena (Z 700P) atomic absorption spectrophotometer graphite furnace method was used to determine the concentration of Cd and Pb in different parts of rice. The determination of zinc in brown rice was conducted by (iCAP 7000 Series) inductively coupled plasma emission spectrometry. Nitrogen and phosphorus concentration of brown rice were determined by an AA3 continuous flow analyzer (SEALXY-2SAMPLER) and the potassium content was determined by a flame photometer (Sherwood 410). The total Cd and Pb concentration in the soil samples was determined by graphite furnace atomic absorption spectrophotometry according to GB/T 17141-1997 in China. The determination of DTPA-Cd and DTPA-Pb in soil according to GB/T 23739-2009 in China was made by the Jena (Z 700P) atomic absorption spectrophotometer.

Soil samples (GBW07461) and plant samples (GBW10045) were used for quality control, and the analysis results were within the allowable error range.

### 2.5. Analysis Method

Calculate relevant indicators according to the following Formula:

$$BCF = \text{certain element concentration in brown rice (mg·kg}^{-1})/\text{soil concentration of this element (mg·kg}^{-1}) \quad (1)$$

$$TF_{grain\text{-}straw} = \text{heavy metal concentration of brown rice (mg·kg}^{-1})/\text{heavy metal concentration of rice straw (mg·kg}^{-1}) \quad (2)$$

$$TF_{straw\text{-}root} = \text{heavy metal concentration of rice straw (mg·kg}^{-1})/\text{heavy metal concentration of rice root (mg·kg}^{-1}) \quad (3)$$

Excel 2016 was used for data collation, and SPPS 23.0 was used for variance analysis and correlation analysis for statistical analysis of test data. The means between treatments were compared by the least significant difference method (LSD). All data in the chart were represented by mean $\pm$ standard deviation (M $\pm$ SD), and the significant difference level was $p < 0.05$. Origin 2017 was used for mapping.

## 3. Results

### 3.1. Effects of Foliar Zn Application on Rice Yield

As shown in Figure 1, leaf spraying with zinc fertilizer increased rice yield between 7.584–7.803 t·hm$^{-2}$, and compared with CK, rice yield under different treatments significantly increased by 4.28–7.29% ($p < 0.05$). The yield increases of T3 (twice spraying) and T4 (three times spraying) were higher than those of T1 and T2 (both once spraying). The difference was significant compared with T1 ($p < 0.05$).

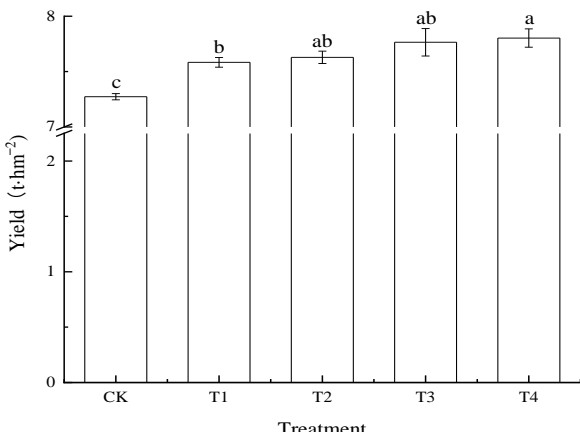

**Figure 1.** Effect of foliar Zn application on rice yield. Note: Different lowercase letters indicate significant differences between treatments ($p < 0.05$). The letters indicating significant differences are marked in the order from large to small, and the maximum average number is marked as a.

*3.2. Effects of Foliar Zn Application on Cd and Pb Concentration of Rice*

3.2.1. Effects of Foliar Zn Application on Cd and Pb Concentration in Brown Rice

As shown in Figure 2, leaf spraying zinc fertilizer can make Cd concentration of brown rice range from 0.183–0.294 mg·kg$^{-1}$, which is significantly lower than CK by 29.52–56.01% ($p < 0.05$). Among them, T3 (sprayed twice) and T4 (sprayed three times) had significant differences compared with T1 and T2 (both sprayed once) ($p < 0.05$), and the Cd concentration of brown rice under T3 and T4 treatment was 0.189 mg·kg$^{-1}$ and 0.183 mg·kg$^{-1}$, respectively. The Cd concentration in brown rice can be reduced to the standard value (0.2 mg·kg$^{-1}$) stipulated in GB 2762-2017. Leaf spraying of zinc fertilizer could reduce plumbum content in brown rice between 0.155 and 0.192 mg·kg$^{-1}$, which was significantly reduced by 11.10–28.34% compared with CK ($p < 0.05$). T4 had the best effect on Pb concentration in brown rice, and the difference was significant compared with other treatments ($p < 0.05$). However, all treatments could reduce the Pb concentration in brown rice to below the standard value of GB2762-2017 (0.2 mg·kg$^{-1}$).

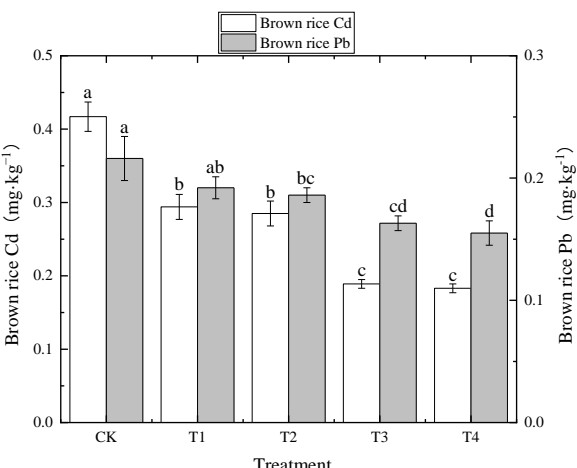

**Figure 2.** Effect of foliar Zn application on Cd and Pb concentration in brown rice. The letters indicating significant differences are marked in the order from large to small, and the maximum average number is marked as a.

3.2.2. Effects of Foliar Zn Application on Cd Concentration of Rice Plants

As can be seen from Figure 3, Cd concentration in rice husks under leaf spraying zinc fertilizer ranged from 0.093–0.118 mg·kg$^{-1}$, and was significantly decreased by 23.22–39.14% compared with CK ($p < 0.05$). The reduction effect of T3 (two sprays) and T4 (three sprays)

was significantly different from that of T1 and T2 (both one spray) ($p < 0.05$), but there was no significant difference between T3 and T4. The cadmium content of straw ranged from 0.888–0.979 mg·kg$^{-1}$, which decreased by 6.91–15.59% compared with CK. There was no significant difference in the decrease of T1 and T2 compared with CK, but there was a significant difference in the decrease of Cd concentration of straw after two or more spraying sessions (T3 and T4) compared with one spraying session ($p < 0.05$). Cd concentration in the root was 1.069–1.143 mg·kg$^{-1}$, and the decrease was not significant compared with CK ($p < 0.05$).

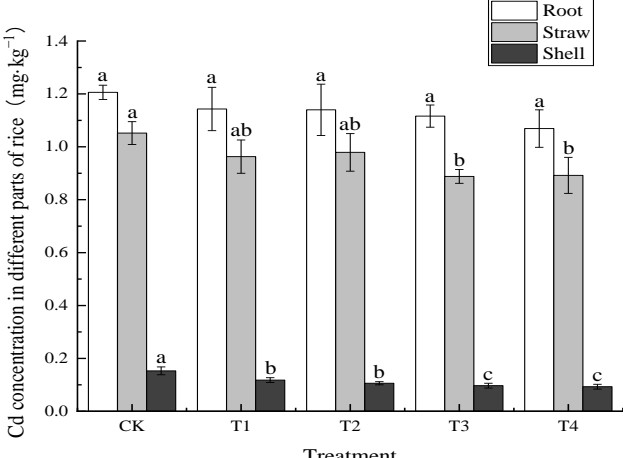

**Figure 3.** Effect of foliar Zn application on Cd concentration in different parts of rice. The letters indicating significant differences are marked in the order from large to small, and the maximum average number is marked as a.

### 3.2.3. Effects of Foliar Zn Application on Pb Concentration of Rice Plants

As shown in Figure 4, the Pb concentration of rice husk under foliar zinc spraying was between 0.214–0.284 mg·kg$^{-1}$, and was significantly decreased by 11.89%–33.54% compared with CK ($p < 0.05$), and the effect of T4 (three times of spraying) was the most significant compared with other treatments ($p < 0.05$). The Pb concentration of straw was between 1.427–2.123 mg·kg$^{-1}$, which was significantly decreased by 24.83%–49.48% compared with CK ($p < 0.05$). The reduction effect of T3 (twice spraying) and T4 (three spraying) was significantly different from that of T1 and T2 (once spraying) ($p < 0.05$). Pb concentration in the roots ranged from 0.931–1.162 mg·kg$^{-1}$. Compared with CK, all treatments significantly reduced Pb concentration in the roots, and there were significant differences between two or more spraying and one spraying sessions ($p < 0.05$).

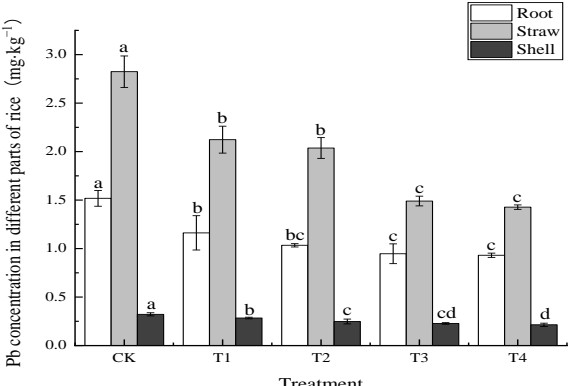

**Figure 4.** Effect of foliar Zn application on Pb concentration in different parts of rice. The letters indicating significant differences are marked in the order from large to small, and the maximum average number is marked as a.

*3.3. Effects of Foliar Zn Application on Cd and Pb Enrichment and Transport in Rice*

3.3.1. Effects of Foliar Zn Application on Cd Enrichment and Transport in Rice

As can be seen from Table 1, there was a significant difference ($p < 0.05$) between 0.121–0.194 in the Cd enrichment capacity of brown rice under foliar zinc application. Compared with CK, the Cd enrichment capacity of brown rice under foliar zinc application decreased by 28.18–55.02%, respectively. The effects of T3 (two sprays) and T4 (three sprays) were significantly different from those of T1 and T2 (both one spray) ($p < 0.05$), but there was no significant difference between T3 and T4. The transport coefficients of rice straw-root, brown rice–straw and brown rice–root ranged from 0.796–0.864, 0.206–0.306 and 0.169–0.257, respectively. Leaf zinc application could significantly reduce the transport coefficients of brown rice–straw and brown rice–root, and the difference between more than two spraying and one spraying session was significant ($p < 0.05$).

**Table 1.** Effects of foliar Zn application on Cd accumulation and transport in rice.

| Treatment | Soil Cd | BCF$_{\text{Brown Rice}}$ | TF$_{\text{Straw-Root}}$ | TF$_{\text{Brown Rice-Straw}}$ | TF$_{\text{Brown Rice-Root}}$ |
|---|---|---|---|---|---|
| CK | 1.545 ± 0.020 a | 0.270 ± 0.015 a | 0.872 ± 0.018 a | 0.396 ± 0.025 a | 0.346 ± 0.020 a |
| T1 | 1.516 ± 0.030 a | 0.194 ± 0.012 b | 0.844 ± 0.052 a | 0.306 ± 0.033 b | 0.257 ± 0.021 b |
| T2 | 1.509 ± 0.024 a | 0.189 ± 0.012 b | 0.864 ± 0.114 a | 0.293 ± 0.037 b | 0.251 ± 0.022 b |
| T3 | 1.539 ± 0.016 a | 0.122 ± 0.005 c | 0.796 ± 0.032 a | 0.212 ± 0.007 c | 0.169 ± 0.011 c |
| T4 | 1.518 ± 0.061 a | 0.121 ± 0.010 c | 0.839 ± 0.111 a | 0.206 ± 0.008 c | 0.172 ± 0.016 c |

Note: Different lowercase letters in the same column indicate significant differences among treatments ($p < 0.05$). The same below.

3.3.2. Effects of Foliar Zn Application on Pb Enrichment and Transport in Rice

As can be seen from Table 2, leaf zinc application has a significant difference in the Pb enrichment ability of brown rice ($p < 0.05$), and the enrichment coefficient is about 0.002, with no significant difference among treatments ($p < 0.05$). The transport coefficients of rice straw–root, brown rice–straw and brown rice–root ranged from 1.532–1.969, 0.091–0.110 and 0.168–0.180, respectively. Leaf zinc application could reduce the transport coefficients of straw–root brown rice–straw and brown rice–root, and the difference between more than two spraying and one spraying session was significant ($p < 0.05$).

**Table 2.** Effects of foliar Zn application on Pb accumulation and transport in rice.

| Treatment | Soil Pb | BCF$_{\text{Brown Rice}}$ | TF$_{\text{Straw-Root}}$ | TF$_{\text{Brown Rice-Straw}}$ | TF$_{\text{Brown Rice-Root}}$ |
|---|---|---|---|---|---|
| CK | 93.54 ± 2.602 a | 0.002 ± 0.001 a | 1.867 ± 0.193 a | 0.077 ± 0.009 c | 0.142 ± 0.005 b |
| T1 | 92.42 ± 1.830 a | 0.002 ± 0.000 a | 1.842 ± 0.159 ab | 0.091 ± 0.009 bc | 0.168 ± 0.030 ab |
| T2 | 91.09 ± 1.361 a | 0.002 ± 0.000 a | 1.969 ± 0.085 a | 0.092 ± 0.008 b | 0.180 ± 0.008 a |
| T3 | 90.74 ± 1.892 a | 0.002 ± 0.000 a | 1.584 ± 0.158 bc | 0.110 ± 0.001 a | 0.174 ± 0.016 ab |
| T4 | 90.70 ± 2.891 a | 0.002 ± 0.000 a | 1.532 ± 0.020 c | 0.109 ± 0.007 a | 0.166 ± 0.009 ab |

*3.4. Effects of Foliar Zn Application on Zinc, Nitrogen, Phosphorus and Potassium Concentrations in Brown Rice*

As can be seen from Figure 5, leaf zinc application made the zinc content in brown rice range from 18.16–20.68 mg·kg$^{-1}$, which was significantly increased compared with CK ($p < 0.05$). T4 (three times of spraying) increased the zinc concentration of brown rice by 34.64%, and T3 (two times of spraying) and T2 (one time of spraying) increased the zinc content of brown rice between 22.14% and 29.60%, which was significantly higher than CK and T1. Leaf zinc application could increase nitrogen, phosphorus and potassium content in brown rice by 11.46–37.54%, 7.83–21.10% and 15.49–33.03%, respectively. The effect of T3 and T4 treatment was significantly better than that of CK, T1 and T2 ($p < 0.05$).

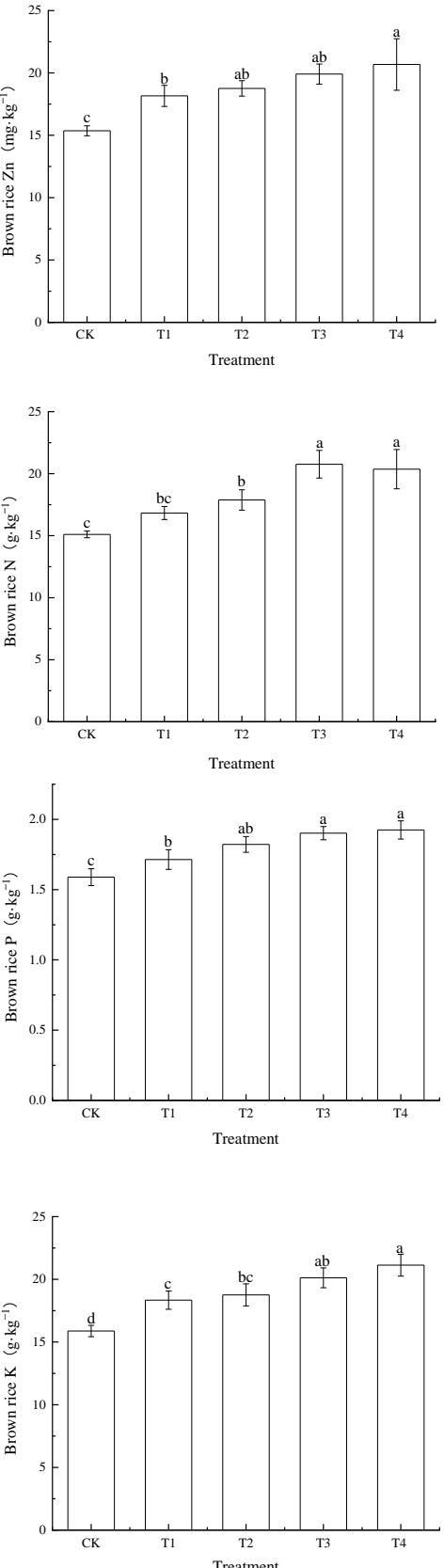

**Figure 5.** Effect of foliar Zn application on the concentration of Zn, N, P and K in brown rice. The letters indicating significant differences are marked in the order from large to small, and the maximum average number is marked as a.

### 3.5. Influence of Foliar Zn Application on Economic Benefits

The leaf control materials involved in this experiment, i.e., rice prices, seeds, fertilizers, machinery, labor and other costs, were obtained by combining market investigation, and the specific prices were subject to actual market prices. As can be seen from Figure 2, only one application of leaf zinc fertilizer at the heading stage or filling stage cannot simultaneously reduce the heavy metal Cd and Pb concentration in brown rice to below the standard value of GB 2762-2017 (0.2 mg·kg$^{-1}$). Therefore, the unit price of rice is converted to 60% of the market price. On the basis of other management levels being consistent, the economic benefit pair is shown in Table 3. The input-output of T3 and T4 treatments was higher, which were 1.863 and 1.862, respectively. The difference between spraying zinc fertilizer twice or more and spraying zinc fertilizer once was significant ($p < 0.05$).

**Table 3.** Effect of foliar Zn application on economic benefits.

| Treatment | Input (¥·hm$^{-2}$) | | | Total Input (¥·hm$^{-2}$) | Total Output (¥·hm$^{-2}$) | Input–Output Ratio |
|---|---|---|---|---|---|---|
| | **Blade Material** | **Fertilizer** | **Other** | | | |
| CK | 0 | 1215 | 9500 | 10,715 | 11,345 ± 43 c | 1.059 ± 0.004 c |
| T1 | 60 | 1215 | 9500 | 10,775 | 11,831 ± 68 b | 1.098 ± 0.006 b |
| T2 | 60 | 1215 | 9500 | 10,775 | 11,902 ± 88 b | 1.105 ± 0.008 b |
| T3 | 120 | 1215 | 9500 | 10,835 | 20,189 ± 322 a | 1.863 ± 0.030 a |
| T4 | 180 | 1215 | 9500 | 10,895 | 20,286 ± 215 a | 1.862 ± 0.020 a |

Note: The unit price of rice is 2.6 CNY/kg$^{-1}$. If heavy metals exceed the standard, the unit price of rice is calculated as 60%. Other costs: including seeds, pesticides, machinery, labor, etc.

## 4. Discussion

### 4.1. Effects of Leaf Zinc Application at Different Periods on Reducing Cd and Pb Accumulation in Rice

Exogenous zinc enhanced the competition between Zn and Cd in plants and reduced the transport and accumulation of cadmium. Studies have shown that when the degree of Cd pollution is low, zinc and cadmium show an antagonistic relationship, but when the degree of Cd pollution is high, Zn and Cd show a synergistic relationship or no interaction [34]. Previous studies have shown that leaf Zn application can reduce the concentration of heavy metals in edible parts of rice [14,35], wheat [36–38], Chinese cabbage [39] and cucumber [40]. These may be that leaf application of Zn can cause the affinity membrane transporters shared by Zn and Cd in rice leaves to produce Zn-Cd antagonism, thus reducing the absorption of Cd by rice. Meanwhile, Zn can compete with Cd for binding sites on cells, ultimately achieving the purpose of reducing Cd concentration. In this study, leaf spraying of Zn fertilizer significantly reduced the concentrations of Cd and Pb in brown rice, by 29.52–56.01% and 11.10–28.34%, respectively, compared with CK, and the concentrations of Cd and Pb in brown rice showed a decreasing trend with the increase of spraying times, but the decline slowed down after reaching a certain number of times. Wang et al. [5] found that leaf spraying Zn fertilizer at the early stage of rice filling could effectively reduce the concentration of Cd in grains. In this study, the effect of leaf Zn fertilizer spraying at the rice filling stage was better than that at tillering stage and heading stage to reduce the concentration of Cd and Pb in brown rice. Previous studies have shown that Cd and Zn use the same transporter for absorption and transport in plants [41,42]. Zn competes with Cd for the heavy metal binding site on this transporter and inhibits the transmembrane transport of Cd [20], thus changing the transport efficiency of these two heavy metals from root to stem [43]. Therefore, increasing the concentration of Zn leads to a decrease in the concentration of Cd. Tian et al. [9] found that spraying Zn fertilizer at the leaf surface during the rice filling stage could increase the transfer of Zn to grains and reduce the accumulation of Cd. Han et al. [14] found that foliar Zn application could effectively reduce the concentration of heavy metals in grains by promoting Zn allocation and inhibiting the transport of heavy metals from vegetative organs to grains. Previous

studies [23,44] showed that the main reason to control the concentration of heavy metals in seeds was to reduce the concentration of heavy metals in flag leaves by spraying Zn on the leaf surface. Flag leaf and Section 1 have strong Cd enrichment ability and play an important role in the transfer of Cd to grain [45,46]. Therefore, it is an important way to ensure the safety of rice to reduce the absorption and accumulation of Cd and Pb by Zn fertilizer in leaf surface.

*4.2. Effects of Leaf Zinc Application at Different Periods on Rice Yield and Zn-Rich Effect of Brown Rice*

Zn is mainly involved in key physiological processes such as chlorophyll synthesis, photosynthesis and respiration in crops. Zn deficiency in crops can lead to leaf degreenization, macular appearance in leaf veins, small leaf deformation and slow growth [47]. Zn is highly mobile in crop phloem [48]. Zn sprayed on the leaf surface can be absorbed by the leaf epidermis and transported to other parts of the plant through xylem and phloem. Migration from nutrient tissue to grain is an important pathway for zinc accumulation in grain. Rice is a crop sensitive to Zn. Application of Zn fertilizer can significantly increase Zn concentration, relieve Zn deficiency and increase yield. In this study, leaf spraying Zn fertilizer can increase rice yield, and the yield increase rate of each treatment is 4.28%–7.29% compared with blank. Zn remobilization in aging leaf tissues plays a key role in Zn accumulation in wheat grains because of the NAM-B1 transcription factor [49]. Li et al. [50] believed that the application of leaf Zn fertilizer at the early stage of filling could make the Zn in crop nutrient tissue transfer back to the grains, thus producing high-quality crops. Cakmak et al. [51] study showed that, compared with early application of Zn, spraying zinc fertilizer on the leaf surface at the later growth stage of crops can significantly increase the grain Zn concentration. Ozturk et al. [33] found that high Zn accumulation appeared in the early stage of grain formation, such as the early stage of filling, and spraying 10 times was better than spraying 3 times under the same amount of conditions. In this study, the increase of Zn concentration in brown rice by spraying Zn fertilizer on leaf surface at the filling stage was higher than that at heading stage, and the Zn concentration increased with the increase of spraying times. The Zn spraying period is an important factor affecting Zn concentration in grains. After applying 0.4% $ZnSO_4$ to leaves, the Zn concentration in the grains, leaves and roots of rice was significantly increased. Zn adsorbed on leaves is transported to the grain through phloem [51]. In addition, other studies have found that Zn uptake by soil and roots increases after leaf surface Zn application, and then Zn is transported from roots to grains, finally increasing the Zn concentration in rice grains. The high concentration of Zn in rice grains indicates that Zn spraying on the leaf surface can play a role of biological strengthening and improve the quality of rice grains. In this study, leaf spraying with zinc fertilizer at different periods could increase Zn concentration in brown rice, which increased by 22.14–34.64% compared with CK. Therefore, in the growth and development stage of rice, the application of Zn to the nutrient tissue can promote the accumulation of Zn in grain and increase the concentration of Zn in grain, which is a good way to alleviate the Zn deficiency in the human body. Improving the nutritional quality of grain has a significant impact on human health and happiness.

## 5. Conclusions

Spraying Zn fertilizer at the growth stage of rice can increase the yield of rice compared with CK, and the effect of spraying Zn fertilizer twice or more is better. Leaf surface control of Zn fertilizer can significantly reduce the Cd and Pb concentration in brown rice, and the Cd and Pb concentration in brown rice can meet the Chinese standard GB 2762-2022 after two or more spraying. At the same time, the Cd enrichment ability of rice plant parts can be reduced, and the Cd and Pb concentration in brown rice can be reduced by reducing the transport ability of Cd and Pb in rice roots to straw parts and then to brown rice. Moreover, it can significantly increase the Zn concentration in brown rice, and the Zn enrichment effect

of spraying three times is the most significant. The nitrogen, phosphorus and potassium concentration in brown rice will also increase with the increase of spraying times.

**Author Contributions:** Methodology, C.C.; Formal analysis, J.X.; Investigation, Y.G.; Data curation, H.H.; Writing—original draft, J.H.; Writing—review and editing, R.T.; Supervision, Y.M.; Project administration, Z.M. All authors have read and agreed to the published version of the manuscript.

**Funding:** The National Key R&D Program Project "Technology Demonstration of Cadmium Arsenic Pollution Prevention and Ecological Security in Urban Rural Areas" (No.2018YFD0800203) and the Anhui Provincial Science and Technology Major Research Project "Development and Application of Efficient Nanoremediation Materials for Heavy Metal Pollution in Farmland" (No. 17030701053).

**Institutional Review Board Statement:** Not applicable.

**Informed Consent Statement:** Informed consent was obtained from all subjects involved in the study.

**Data Availability Statement:** The data presented in this study are available on request from the corresponding author.

**Conflicts of Interest:** The authors declare no conflict of interest.

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
