# Peer review of "Effect of Leaf Surface Regulation of Zinc Fertilizer on Absorption of Cadmium, Plumbum and Zinc in Rice (Oryza sativa L.)"

_sustainability, doi:10.3390/su15031877_

Round 1

Reviewer 1 Report

This study provided a study on zinc foliage fertilizer on absorption of Cd, pb and Zinc in Rice (Oryza sativa L.) in Cd and Pb combined polluted farmland. Although many studied have been reported on Si foliage fertilizer on cadmium, fewer studied were conducted with a field trial on the chromium and Plumbum combined pollution . However, there are many flaws in this MS. At present, it should be not accepted before the authors corrected them.

Comments and Suggestions for Authors

1- Please reduce your article abstract section. 

2- Please revise your article key words, choose suitable words.

3- -Please clearly mention your objectives and novelty. 

4- please remove numbering from your conclusion section. 

5- Please revise your discussion concisely

6-Please supplement acknowledgments on the funding and assistance for the project.

7-Please improve the language of the current version. 

Author Response

  • Please reduce your article abstract section. 

Dear reviewer, according to your suggestion, the author has simplified the abstract.

  • Please revise your article key words, choose suitable words.

Dear reviewer, according to your suggestions, the author has modified the key words of the article.

  • -Please clearly mention your objectives and novelty. 

Dear reviewers, according to your suggestions, the author has revised the objectives and novelty of this study.

  • please remove numbering from your conclusion section. 

Dear expert reviewer, according to your suggestion, the author has deleted the number in the conclusion and simplified the conclusion part.

  • Please revise your discussion concisely

Dear reviewer, the author has simplified the discussion section of the article according to your suggestion.

  • Please supplement acknowledgments on the funding and assistance for the project.

Dear reviewer, according to your suggestion, the author adds this part of content.

  • Please improve the language of the current version. 

Dear reviewer, according to your suggestions, the author has modified the possible English expressions in the article.

Reviewer 2 Report

Reviewed manuscript “Effect of Leaf Surface Regulation of Zinc Fertilizer on Absorp-
tion of Cadmium, Plumbum and Zinc in Rice (Oryza sativa L.)
is an original and interesting study. Authors comprehensively demonstrated the application effect of Zinc Fertilizer on Absorp-tion of Cadmium, Plumbum and Zinc in Rice (Oryza sativa L.). Study is very interesting and it would add scientific contribution in literature. Authors collected lot of data with strong reasoning. I would suggest minor revision.

Following are some suggestions for further improvements:

First few lines of the abstract should be about the importance of study.

Lines 45-50: Consider revising the lines.

Revise the reference style 3 at line 52.

Lines 80-82: Consider revising the lines.

Lines 118-127: Consider revising the lines.

Quality of figures is very low. It should be uniform in style.

Introduction and Discussion section needs to further strengthen by latest studies on the subject. Improve the discussion section. Please summarize the conclusion section which is very large.

At some places in the text, there are grammatical mistakes that need to be corrected by some native English colleague.

To further strengthen introduction, following latest studies etc. are suggested to cite for the application effect of Zinc Fertilizer on Absorp-tion of Cadmium, Plumbum, Zinc and other heavy metals in agronomic crops.

Author Response

First few lines of the abstract should be about the importance of study.

Dear reviewer, according to your suggestion, the author has modified and simplified part of the abstract, please check.

Lines 45-50: Consider revising the lines.

Dear reviewer, according to your suggestions, the author has modified these contents and marked them in red. Please refer to them.

Revise the reference style 3 at line 52.

Dear reviewer, according to your suggestions, the author has modified these contents and marked them in red.

Lines 80-82: Consider revising the lines.

Dear reviewer, according to your suggestions, the author has modified these contents and marked them in red. Please refer to them.

Lines 118-127: Consider revising the lines.

Dear reviewer, according to your suggestions, the author has modified these contents and marked them in red. Please refer to them.

Quality of figures is very low. It should be uniform in style.

Dear reviewer, the author has adjusted this part of content to make it consistent in format, please refer to it.

Introduction and Discussion section needs to further strengthen by latest studies on the subject. Improve the discussion section.

Dear reviewer, according to your suggestion, the author has revised the preface and discussion part, which have been marked in red in the paper, please refer to it.

Please summarize the conclusion section which is very large.

Dear reviewer, according to your suggestions, the author has simplified the conclusion to make it more concise.

At some places in the text, there are grammatical mistakes that need to be corrected by some native English colleague.To further strengthen introduction, following latest studies etc. are suggested to cite for the application effect of Zinc Fertilizer on Absorp-tion of Cadmium, Plumbum, Zinc and other heavy metals in agronomic crops.

Dear reviewer, according to your suggestions, the author has adjusted the relevant content of the article.
